# Investigation of Fluidity and Strength of Enhanced Foam-Cemented Paste Backfill

**DOI:** 10.3390/ma15207101

**Published:** 2022-10-13

**Authors:** Xiuzhi Shi, Zhengkai Zhao, Xin Chen, Kun Kong, Jingjing Yuan

**Affiliations:** 1School of Resources and Safety Engineering, Central South University (CSU), Changsha 410083, China; 2Department of Endocrinology, The Third Xiangya Hospital, Centre South University, Changsha 410013, China

**Keywords:** foam-cemented paste backfill, foam content, fluidity, unconfined compressive strength, scanning electron microscopy

## Abstract

To solve the problems of high cement dosage and poor fluidity of conventional cemented paste backfill (CPB) materials, the fluidity and strength properties of foam-cemented paste backfill (FCPB) were studied in combination. Based on determining the optimum contents of a foaming agent and a foam stabilizer, FCPB density was measured. To investigate the fluidity and strength of FCPB under different foam contents (0%, 5%, 10%, 15%, 20%, 25%, 30%, and 40%), different solid contents (75 wt.% and 77 wt.%), and different cement-tailing ratios (1:4 and 1:5), spread tests and unconfined compressive strength (UCS) tests were conducted. In addition, the FCPB microstructure was analyzed by scanning electron microscopy (SEM). The results indicate that the optimum combination dosages of sodium lauryl sulfate (K12) and sodium carboxymethyl cellulose (CMC) are 0.5 g/L and 0.2 g/L. The density decreases with the foam content (FC), but the fluidity and strength of the FCPB increase first and then decrease with the FC. In addition, the microstructure analysis explains the enhanced strength of FCPB by adding foam. These results contribute to further understanding the effect of foam content on the fluidity and strength of the FCPB.

## 1. Introduction

In mining operations, cemented paste backfill (CPB) is playing an increasingly important role in the mine backfill [1,2,3]. CPB is used in numerous countries because of its techno-economic advantages and higher utilization of industrial waste compared to other backfill technologies [4,5,6]. CPB not only saves drainage costs and reduces pollution by eliminating the need for dewatering but also ensures the strength of the backfill body. However, CPB slurry has generally difficulty achieving self-flowing transport because of its large yield shear stress and plastic viscosity, and a high-pressure pump system is required to assist in its transport [1]. In addition, this system involves a large investment in equipment and has high technical requirements, making the backfill costly. Therefore, researchers have been paying a lot of attention to improving the transport properties of CPB slurry while satisfying the strength requirements of the backfill [7]. They have explored the effect of admixtures on the rheological and mechanical properties of CPB. For example, materials such as fly ash, ground granulated blast-furnace slag, and high-performance fiber materials have been used to increase the strength of CPB samples [8,9,10,11,12], and additives such as red mud, superplasticizers, and sulfate can improve the rheological properties of CPB slurry [13,14,15].

With the successful application of foam concrete in several fields [16,17,18,19], foaming agents were mixed into the backfill slurry to overcome the shortcomings associated with conventional mine backfill materials, thus developing a new backfill technique, referred to as foam mine backfill. Some claim that foam mine backfill materials have many advantages, such as reduced weight, improved rheology, and minimized cost, so they have the potential for application in underground mines [20]. Therefore, researchers began to add foaming agents to CPB materials and investigate the effect of foaming agent content on the physical and mechanical properties of foam-cemented paste backfill (FCPB). Xu et al. [21] explored the effect of solid content (SC), cement-tailing ratio (CTR), curing time (T), and foaming agent content on the unconfined compressive strength (UCS) of the FCPB. They found that the UCS decreased with an increase in the foaming agent content. Qiu et al. [22] also observed a significant negative correlation of the UCS of the FCPB with the foaming agent content. However, these studies analyzed the effect of foaming agent content on the UCS of the FCPB (although only unilaterally), ignoring the effect of foaming agent content on the fluidity of the FCPB. Zhang et al. [23] evaluated the influence of aggregate gradation and foaming agent (hydrogen peroxide) content on FCPB fluidity, strength, and microstructure. They reported that the hydrogen peroxide content had a negative effect on the fluidity and strength of the FCPB slurry. Xu et al. [24] also observed that the fluidity and UCS of the FCPB decreased approximately linearly as the hydrogen peroxide content increased. Hefni and Hassani [25] investigated the relationship between the UCS of the foam mine backfill and the binder content, the amount of entrained air, and the foam mixing time. They found that the UCS of the foam mine backfill was negatively correlated with the amount of entrained air but the foaming agent appeared to have a plasticizing effect, which can help to improve the fluidity of the mine backfill slurry and facilitate its transport through pipelines.

The conclusions in terms of strength are largely consistent, i.e., FCPB strength decreases with an increase in the foaming agent content or the foam content. It is widely accepted that the pore structure significantly affects the backfill strength [26,27,28]. The more the foam, the greater the porosity (especially the number of large pores) of the FCPB samples, resulting in lower FCPB strength. However, Xu et al. [24] reported that with an increasing foam stabilizer content, the UCS of the FCPB increased first and then decreased. The foam stabilizer decreased the number of connected pores in the FCPB, which improved the UCS of the FCPB. Therefore, the passive influence of foaming agents on FCPB strength may be weakened by adding the foam stabilizer. The difference is that the conclusions on FCPB fluidity are controversial, which may be related to the form of foam added. Hefni et al. added premade foam to the backfill slurry, while other researchers added the foaming agent directly to the backfill slurry. The amount of foaming agent added directly is small, but the foam content in the slurry may be high. According to water film thickness theory, the foam increases the spacing among the solid particles and so the void-filling water increases, decreasing the water film thickness [29]. Therefore, the effect of lower levels of foam content on FCPB fluidity should be further investigated.

In this paper, to solve the problems of high cement dosage and poor fluidity of conventional CPB materials, the mechanical, fluidity, and strength properties of the FCPB were further investigated. The optimum contents of the foaming agent and foam stabilizer were determined by studying the foaming ratio and the half-life period. The effects of the foam content (FC) on the density, fluidity, and strength of the FCPB were analyzed, and FCPB microstructure was investigated by scanning electron microscopy (SEM).

## 2. Materials and Methods

### 2.1. Materials

In this paper, the tailings were obtained from a lead-zinc mine in the Guangdong Province of China. The Changsha Research Institute of Mining and Metallurgy analyzed the physical properties and chemical compositions of the tailings based on standards for geotechnical testing method (GB/T 50123-2019) [30]. The particle size distribution of the tailings was determined using a laser particle size analyzer LS13320. The binder used was Portland cement (P.O 42.5), and tap water was used to prepare the FCPB samples.

### 2.2. Foam Preparation

#### 2.2.1. Foaming Agent Content

To date, anionic foaming agents are widely used because of their large foaming ratio, high yield, and low-cost benefits. Studies show that a foaming agent with a carbon chain length of 12 can have the best foaming effect [31]. Zhi et al. [32] compared the foaming performance and cost of K12, LAS, AOS, AEO, and AES and concluded that K12 is the most suitable foaming agent. K12 (sodium lauryl sulfate) is in the form of a white or light-yellow powder, and its structural formula is CH_3_(CH_2_)_11_OSO_3_Na. In this paper, K12 was selected as the foaming agent.

The foaming ratio is the ratio of foam volume to the aqueous solution, which can be used to evaluate the foaming capacity of the foaming agent and select the best foaming agent content. Figure 1 shows the manufacturing procedure of foam. Different weights of K12 (i.e., 0.1 g, 0.2 g, 0.3 g, 0.4 g, 0.5 g, 0.6 g, 0.7 g, and 0.8 g) were dissolved in a certain amount of water, and then water was added to make the solution volume reach 100 mL. The solutions of 1 g/L, 2 g/L, 3 g/L, 4 g/L, 5 g/L, 6 g/L, 7 g/L, and 8 g/L were respectively prepared. The resulting solution was stirred at 1500 rpm for 3 minutes. After full foaming, the foam was poured into a measuring cylinder, and the foam volume was quickly read.

#### 2.2.2. Foam Stabilizer Content

The foaming ratio of the foaming agent is an important indicator to measure the quantity of foam, and the stability of the foam is one of the indicators to evaluate its quality. The foam prepared by the foaming agent cannot remain stable for a long time, so it is necessary to add a foam stabilizer to improve the performance of the foam. Research shows that sodium carboxymethyl cellulose (CMC) has the best foam stabilization effect on K12 compared with sodium chloride, polyacrylamide, polyacrylamide, and hydroxyethyl cellulose [32]. Therefore, in this paper, CMC was selected as the foam stabilizer.

After determining the optimal foaming agent content, a certain weight of K12 was dissolved in a certain amount of water. Different weights of CMC (i.e., 0.05 g, 0.1 g, 0.15 g, 0.2 g, and 0.3 g) were added to the foaming agent solution and water was added to make the solution volume reach 100 ml. 0.5 g/L, 1 g/L, 1.5 g/L, 2 g/L, and 3 g/L of foam stabilizer solutions were respectively prepared. The same method was used to read the volume of the foam. The half-life period of the foam with different foam stabilizer contents was recorded using a stopwatch. The half-life period is the time required for the foam to precipitate half of the volume of the original solution.

### 2.3. FCPB Preparation

Figure 2 shows the manufacturing procedure of FCPB. After the contents of the required materials were weighed, the tailing, cement, and tap water were stirred for 3 min. At the same time, optimum amounts of the foaming agent and the foam stabilizer were mixed and stirred at 3000 rpm for 3 min. Then, the foam was proportionally poured into the cemented paste slurry and slowly stirred to obtain FCPB slurry. Finally, the FCPB slurry was poured into a 70.7 mm × 70.7 mm × 70 mm standard mold to prepare the FCPB sample. The FCPB samples were stored in a curing room at a temperature of about 22 °C and a humidity of about 92% and then maintained for 3, 7, and 28 days.

Table 1 shows the experimental design. Solid content is determined according to the field engineering application. It is generally believed that the FCPB strength increases with an increasing solid content, but the high solid concentration is not conducive to slurry transport. Therefore, the solid content levels are determined based on the ratio of specific backfill material engineering applications in this paper. All three scenarios involve 3 curing times (i.e., 3, 7, and 28 days) and 8 foam contents (i.e., 0%, 5%, 10%, 15%, 20%, 25%, 30%, and 40%), for a total of 72 ratios. Foam content is the volume ratio between the foam and the slurry without the foam. Since the foam contains water, the water brought in by adding foam is subtracted in advance in the calculation of the program raw material dosage to ensure that the solid content remains unchanged.

### 2.4. Test Methods

#### 2.4.1. Fresh Density

Density, as an important indicator of the physical properties of the backfill slurry, can reflect the strength of the backfill body. The fresh FCPB slurry was poured into the measuring cup. Then the volume was read and weighed. The density of the FCPB slurry was calculated from the weight and volume, as in Equation (1):(1)$$\rho =\frac{W}{V}$$
where *ρ* is the density of the slurry, *W* is the weight of the slurry, and *V* is the volume of the slurry.

#### 2.4.2. Spread Test

The spread test evaluated the slurry fluidity by measuring the diameter of the slurry after it had sunk in the conventional slump test (as shown in Figure 3). In this paper, the spread bucket had top and bottom diameters of 100 mm each and a height of 80 mm. The FCPB slurry was placed in the mould and tamped, and then the mould was lifted vertically. After the FCPB slurry stopped flowing, its spread diameters in two perpendicular directions were read. The average spread diameter was considered as the fluidity of the FCPB slurry.

#### 2.4.3. UCS Test

For each of the 72 ratios, 3 identical FCPB samples were prepared, i.e., a total of 216 FCPB samples. The UCS of FCPB samples was measured in a 300 kN capacity TYE-300 type pressure compression testing machine at a loading rate of 0.5 mm/min. The average UCS of three identical FCPB samples was considered the strength. Figure 4 shows the procedure of the UCS test.

#### 2.4.4. SEM Analysis

A scanning electron microscope (SEM) was recommended by ASTM for microstructural analysis of concrete and its similar materials. In this paper, a Quanta FEG 250 field-emission electron microscope with an accelerating voltage of 20 kV was used to observe the complete FCPB samples and analyze the effect of foam on the microstructure properties of FCPB samples.

## 3. Results and Discussion

### 3.1. Particle Size Distribution

Table 2 and Table 3 summarize physical properties and chemical compositions of the tailings, respectively. Figure 5 shows the particle size distribution of the tailings, which was determined using laser particle size analysis technology. As can be seen, d10, d30, d50, and d60 were 8.390 μm, 28.511 μm, 57.085 μm, and 79.519 μm respectively, and the proportion of tailings of <20 µm was only 21.6%. The non-uniformity coefficient (Cu) of the tailing particles was 9.478 and the coefficient of curvature (Cc) was 1.218, indicating that the particle size gradation of tailings is good.

### 3.2. Determination of Admixture Contents

#### 3.2.1. Foaming Agent Content

As shown in Figure 6, the foaming ratio increased from 3.40 to 11.50 when the foaming agent (K12) content increased from 1 g/L to 8 g/L. However, once the foam content exceeded 5 g/L, the foaming ratio no longer increased significantly with foaming agent content. The foaming ratio eventually became a constant value, which means the foaming ability of the foaming agent reached its peak. Considering the cost of the foaming agent, the K12 content selected for the subsequent study was 5 g/L.

#### 3.2.2. Foam Stabilizer Content

Figure 7 shows that the foaming ratio decreased from 10.4 to 7.6 when the foam stabilizer content increased from 0 to 3 g/L The foam stabilizer enhances foam quality but increases the intermolecular force that the foaming agent needs to overcome during the stirring process, thereby inhibiting the performance of the foaming agent [32]. Moreover, the half-life period of the foam increased from 404 s to 945 s when the foam stabilizer content increased from 0 to 3 g/L. When CMC was dissolved in the foaming agent solution, the hydrophobic group in the CMC molecule would form a hydrogen bond with the water molecule, which increased the intermolecular space, giving the foam excellent viscosity. The water retention and the thickness of the foam liquid film were improved, and the ability of the foam liquid film to resist gravity and the internal pressure of the foam were enhanced, thus improving the stability of the foam.

Table 4 shows the Min-Max normalization results of the foaming ratio and half-life period for different foam stabilizer content. It indicated that the best balance of foaming ratio and half-life period was achieved when the foam stabilizer content was 2 g/L. Therefore, the CMC content selected for the subsequent study was 2 g/L.

### 3.3. Density

Table 5 shows the density of the FCPB slurry with different foam content (FC), solid content (SC), and cement-tailing ratio (CTR). Figure 8 shows the relationship between the FC and the density of the FCPB slurry and the fitted curve of density as a function of the FC. The fitting used a cubic function, and the R^2^ values of the three ratios were 0.92, 0.98, and 0.97, respectively, indicating a good fitting effect. The density increased with an increase in the solid content and the cement-tailing ratio but decreased with an increasing foam content. The mean density of the three scenarios decreased by 10.5, 9.8, and 7.7 kg/m^3^ for every 1% increase in the FC. This indicated that the stability of the foam material was good. In addition, a stable cavity was formed in the backfill material, which reduced the number of solid materials used and made the backfill body lighter.

### 3.4. Fluidity

#### 3.4.1. Effect of FCs on the UCS of FCPB

Figure 9 shows the relationship between the FC and the fluidity of the FCPB slurry. The fluidity increased first and then decreased with an increasing FC, but the fluidity of the slurry with foam was always higher than that of the slurry without foam. Hou et al. [33] also came to a similar conclusion in a study of the effect of hydrolyzed protein on the fluidity of foam concrete. These results validated the speculation of Hefni et al. [25] but contradicted the findings of other scholars [23,24], which may be related to the type of foaming agent and the way of adding foam. In this paper, instead of adding the foaming agent directly to the slurry, the foam was first prepared and then added to the slurry, which helps control the foam content more accurately, and the added foam was graded more finely.

As shown in Figure 9, the fluidity increased the fastest when the FC increased from 0% to 5%, reaching the maximum between FCs of 15% and 20%, which indicates that adding a small amount of foam can greatly improve the fluidity of the FCPB slurry by reducing the interparticle friction. However, too much foam consumes a large amount of free water content. According to the water film thickness theory, the water film thickness decreases with the decrease in the free water content, which leads to a decrease in the fluidity of the FCPB slurry [23,29]. Therefore, the fluidity of the FCPB slurry gradually decreased after reaching the peak. Furthermore, there was no significant difference in the fluidity of the FCPB slurry between FCs of 30% and 40%, indicating that there may be a limit to the amount of FC that can be added and once this value is exceeded, the fluidity of the FCPB slurry will no longer decrease with an increasing FC [23].

#### 3.4.2. Effect of SCs and CTRs on the UCS of FCPB

Figure 10a shows the fluidity difference between Scenario 1 (SC = 75 wt.%; CTR = 1:4) and Scenario 2 (SC = 77 wt.%; CTR = 1:4). All the fluidity differences were positive regardless of the FC, showing that the fluidity of FCPB slurry decreased with an increasing solid content. The differences in the fluidity of FCPB slurry with foam were less than those of FCPB without foam in all cases, indicating that the foam weakened the effect of solid content on fluidity. Moreover, the fluidity difference was the lowest when the foam content was 20%, implying that the weakening effect was most pronounced in all cases.

Figure 10b shows the fluidity difference between Scenario 3 (SC = 77 wt.%; CTR = 1:5) and Scenario 2 (SC = 77 wt.%; CTR = 1:4). All the fluidity differences were positive regardless of the FC, showing that the fluidity of the FCPB slurry decreased with an increasing CTR. However, the fluidity differences became higher after adding the foam, indicating that the foam enhanced the influence of the cement–tailing ratio on fluidity. As the foam content reached 20%, the difference was the highest and the enhancement effect of foam was maximized.

### 3.5. Unconfined Compressive Strength

#### 3.5.1. Effect of FCs on the UCS of the FCPB

Figure 11 shows the relationship between the FC and the UCS of the FCPB. The UCS increased first and then decreased with an increasing foam content regardless of the solid content, the cement-tailing ratio, and curing time and eventually fell below the UCS of FCPB samples without foam. For example, when the FC increased from 0% to 5%, the UCS of FCPB samples with a solid content of 77 wt.%, a cement-tailing ratio of 1:4, and a curing time of 28 days increased from 8.17 to 10.67 MPa. Then the UCS decreased to 5.9 MPa as the FC increased to 40%. A further survey of FCPB samples cured for 28 days shows that the UCS increased the most when the FC increased from 0% to 5%. The UCS was considered as the initial strength when the FC was 0. The UCS of FCPB samples with higher initial strength was always higher regardless of the foam content. These results differ from previous studies that purported that the UCS of FCPB samples did not invariably decrease with an increasing foam content, which may be related to the foam content level. Further analysis will be made later, in SEM observation. Moreover, as shown in Figure 11, when the cement-tailing ratio was 1:4 and curing time was 28 days, the UCS of FCPB samples with an FC of 5% and a solid content of 75 wt.% was higher than that of FCPB samples with an FC of 0% and a solid content of 77 wt.%, indicating that adding a moderate amount of foam can reduce the SC for the same strength requirement, so the amount of tailing and cement decreased, which helped reduce the backfill cost.

In addition, the FCPB slurry with lower solid content and higher foam content has a lower density as shown in Table 5, thus, the FCPB was a stronger and lighter backfill material compared to CPB materials. The UCS of FCPB samples increased with increasing curing time irrespective of the foam content, the solid content, and the cement-tailing ratio, as shown in Figure 11. The increase in UCS with curing time can be attributed to the hydration products. The hydration product was proportional to the curing time. It can fill the pores and vesicles in the FCPB and bond with the tailings, increasing FCPB strength [21,34].

#### 3.5.2. Effect of SCs and CTRs on the UCS of the FCPB

Figure 12a shows that the UCS of FCPB samples increased with increasing SC when the foam content, the cement-tailing ratio, and curing time were constant. As the SC increased from 75 wt.% to 77 wt.%, the UCS of FCPB samples with an FC of 5% and a CTR of 1:4 increased from 4.1 to 4.35 MPa for 3 days, from 5.17 to 5.65 MPa for 7 days, and from 9.85 to 10.67 MPa for 28 days. The FCPB samples with a higher SC had more cement hydration products, which bonded with more tailing particles, increasing the UCS.

Figure 12b shows that the UCS of FCPB samples increased with an increasing CTR when the foam content, the solid content, and curing time were constant. As the CTR increased from 1:5 to 1:4, the UCS of FCPB samples with an FC of 5% and an SC of 77 wt.% increased from 2.43 to 4.35 MPa for 3 days, from 4.43 to 5.65 MPa for 7 days, and from 7.65 to 10.67 MPa for 28 days. Studies have found that the UCS of the CPB increases with an increasing CTR because more hydration products with a higher CTR reduce the porosity of the CPB, improving the CPB strength [35,36]. This trend did not change with the addition of foam.

### 3.6. SEM Observation

Studies have shown that the foaming agent has no significant effect on the hydration of cement but can change its strength by altering the structural characteristics of pores [22,23]. The microscopic properties of FCPB samples need to be further observed by SEM. Figure 13 presents the SEM images of FCPB samples (SC = 77 wt.%, CTR = 1:5, and T = 28 d) with FCs of 10% and 30%. The scales were 500 μm, 10 μm, and 5 μm, respectively.

As seen in the SEM images at a scale of 5 μm, the internal hydration reaction products of FCPB samples, such as flocculent calcium silicate hydrates (C–S–H) and acicular ettringite (Aft), were abundant. These hydration products were staggered and bonded to the aggregate particles, which increased the density of the FCPB samples. Some irregular pores (such as Pore 1 and Pore 2 in the figure) were formed between the large particles in the interior of the FCPB and the microparticles and cementation products were attached to the large particles. After some foam filling into the pores between the large particles and the C–S–H gels adhered to the surface of the bubbles to form spherical shells, such as Shell 1 and Shell 2 in the figure. The spherical shells replaced the original pores and bonded the pores to aggregates and other hydration products, increasing the friction between the substances. According to thin shell theory [37], when an FCPB sample is subjected to axial pressure, the spherical shells disperse the axial pressure in multiple directions while being compact (as shown in Figure 14). Some transverse pressures counteracted each other to improve the anti-pressure ability of FCPB samples. Most FCPB samples with a small amount of foam were damaged along the top or bottom cracks after UCS tests, pointing to the obvious anti-pressure effect of spherical shells when the amount of foam was small, so the UCS of FCPB samples increased with an increasing FC at the beginning.

The number of pores in FCPB samples with an FC of 30% was much higher compared to that in FCPB samples with an FC of 10%, as seen in the SEM images at a scale of 500 μm. The excessive foam made the porosity so high that the instability of the structure increased, so the anti-pressure effect of spherical shells was no longer significant. As displayed in Figure 14, the FCPB samples with a high amount of foam had more internal pores and more closely spaced large pores, and cracks were more likely to penetrate between the pores rather than extending along the top or the bottom. Therefore, the UCS of FCPB samples increased first and then decreased.

## 4. Conclusions

In this paper, the properties of the foaming agent and the foam stabilizer were analyzed and the mechanical, fluidity, and compactness properties of FCPB were studied by the spread test, the UCS test, and SEM analysis and then discussed. The following conclusions can be drawn:(1)The foaming ratio increases with the foaming agent content. After the foam stabilizer is added, the foaming ratio tends to decrease but the foam half-life tends to increase. The optimum contents of the foaming agent and the stabilizer are 0.5 wt.% and 0.2 wt.%, respectively.(2)The density of the FCPB decreases with an increasing foam content (FC) but increases with an increase in the solid content and the cement-tailing ratio.(3)The fluidity of the FCPB increases first and then decreases with an increasing FC, but it is always higher than the fluidity of the FCPB without foam. The fastest increase in fluidity is observed when the FC increases from 0% to 5%, and the fluidity reaches the maximum between FCs of 15% and 20%. A small amount of foam helps to improve the fluidity of the FCPB, but too much foam reduces the free water in the slurry, which leads to a decrease in the FCPB fluidity.(4)The UCS of the FCPB increases first and then decreases with an increasing FC and becomes the maximum when the FC increases from 0% to 5%, which is related to the change in the pore structure on adding the foam. The gels adhere to the surface of the bubble to form a spherical shell that replaces the original pore, thus improving the anti-pressure ability of the FCPB. However, excessive foam makes the porosity too high, which decreases the UCS. Furthermore, the UCS increases with an increase in the solid content, the cement-tailing ratio, and curing time.

This paper demonstrates that the fluidity and strength of CPB material can be optimized by adding proper amount of foam. FCPB material is a lighter and lower cost backfill material with great potential for development. This work is important for guiding the use of FCPB materials in mines. Our future work will explore more stable foaming agents and foam stabilizers to further reduce costs while improving the fluidity and strength of FCPB.

## Figures and Tables

**Figure 1 materials-15-07101-f001:**
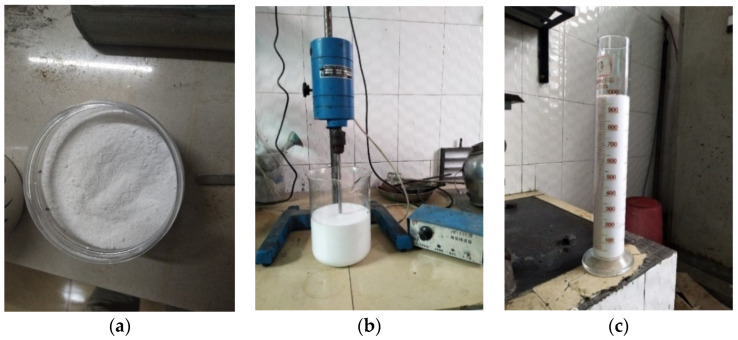
Manufacturing procedure of foam: (**a**) foaming agent; (**b**) stirring fully; (**c**) reading the foam volume.

**Figure 2 materials-15-07101-f002:**
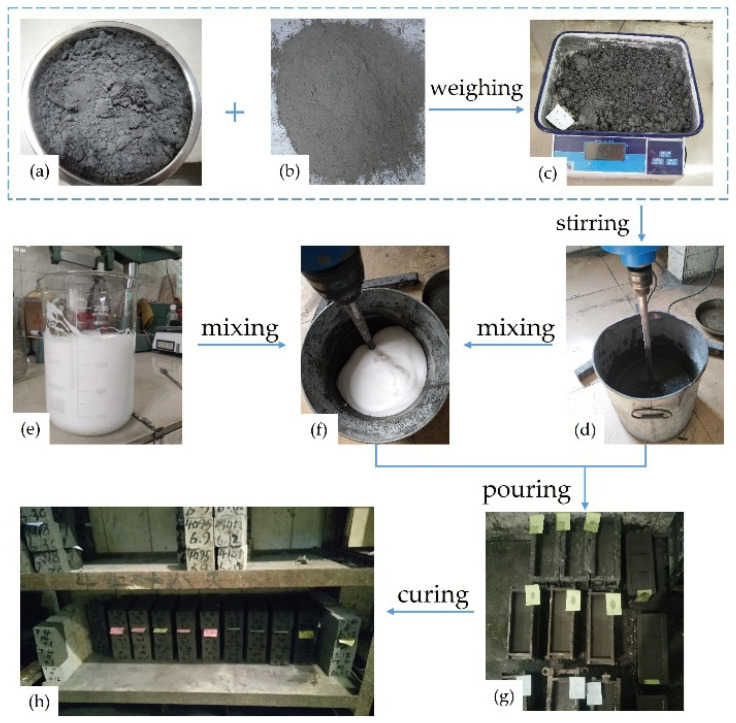
Manufacturing procedure of FCPB: (**a**) tailing; (**b**) cement; (**c**) materials weighing; (**d**) materials stirring; (**e**) foam preparation; (**f**) FCPB slurry preparation; (**g**) FCPB samples preparation; (**h**) curing room.

**Figure 3 materials-15-07101-f003:**
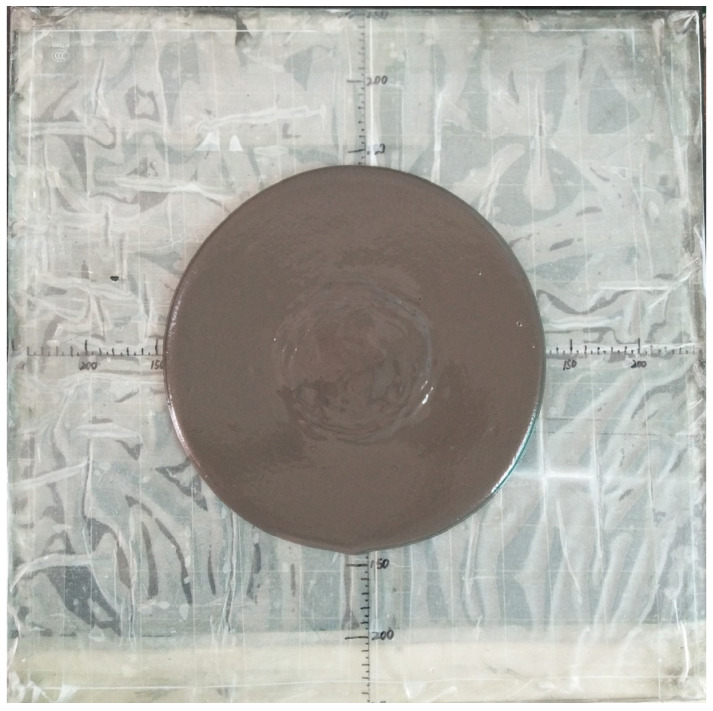
FCPB slurry in the spread test.

**Figure 4 materials-15-07101-f004:**
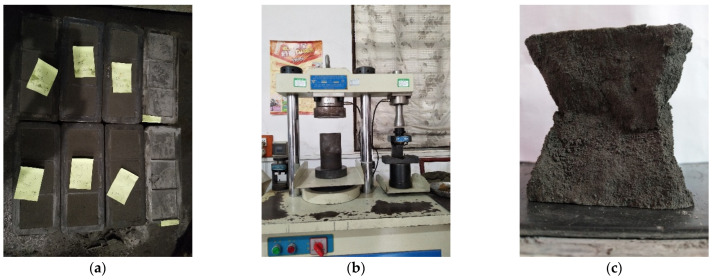
UCS test: (**a**) FCPB samples; (**b**) TYE-300 press machine; and (**c**) FCPB sample after the UCS test.

**Figure 5 materials-15-07101-f005:**
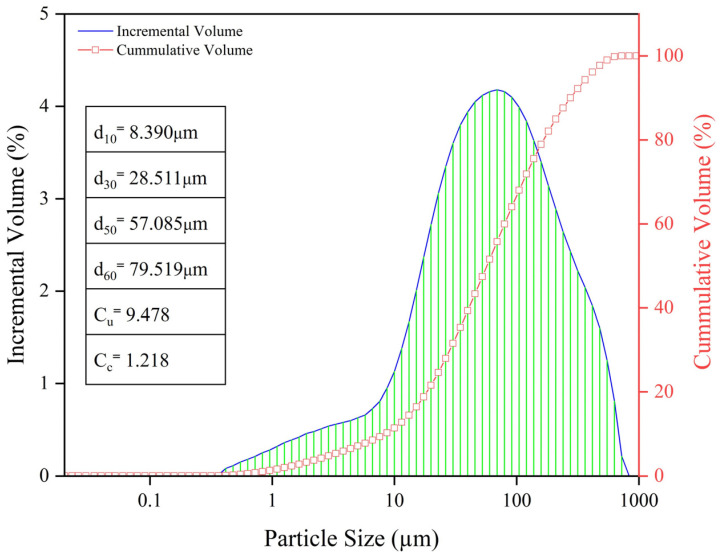
Particle size distribution of tailings.

**Figure 6 materials-15-07101-f006:**
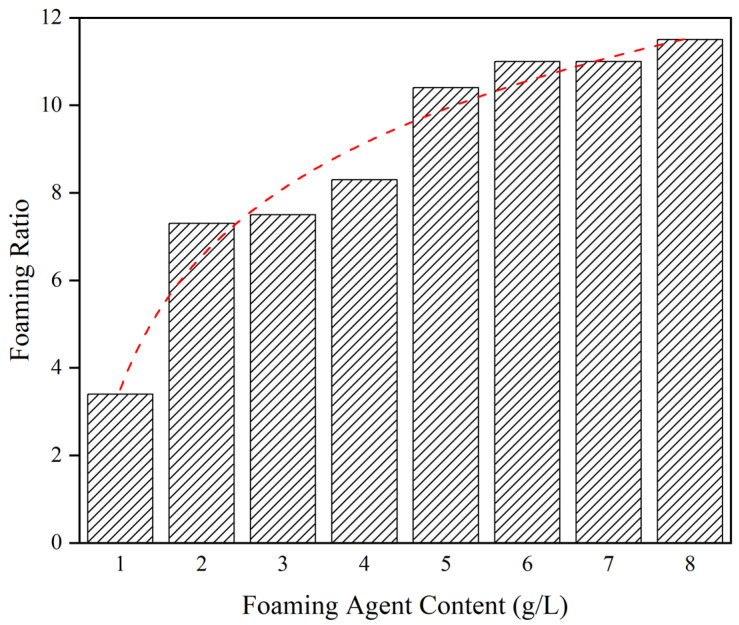
Effect of foaming agent content on foaming ratio.

**Figure 7 materials-15-07101-f007:**
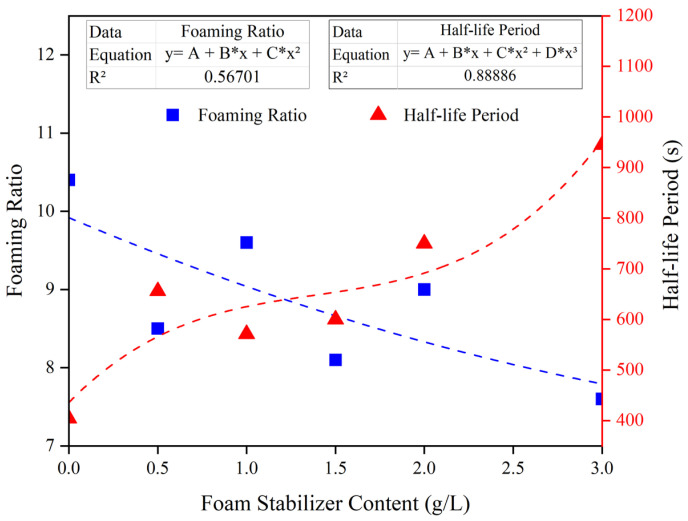
Effect of the foam stabilizer content on the volume and the half-life period of the foam.

**Figure 8 materials-15-07101-f008:**
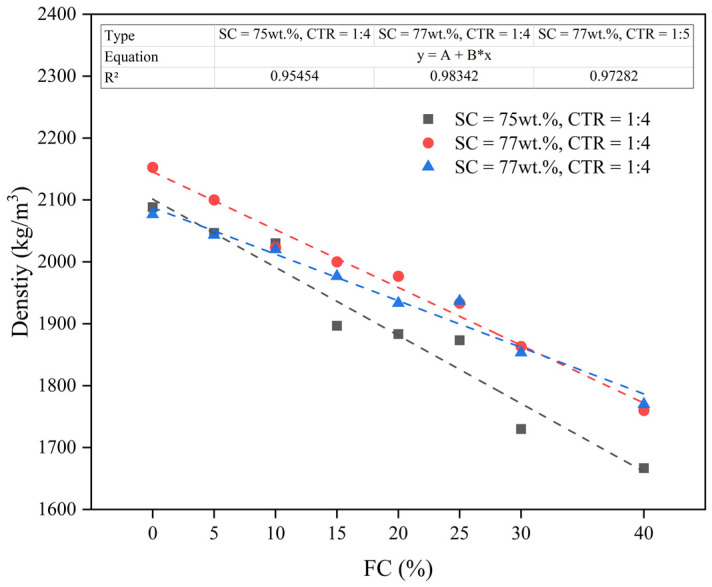
Effect of the FC on the density of the FCPB.

**Figure 9 materials-15-07101-f009:**
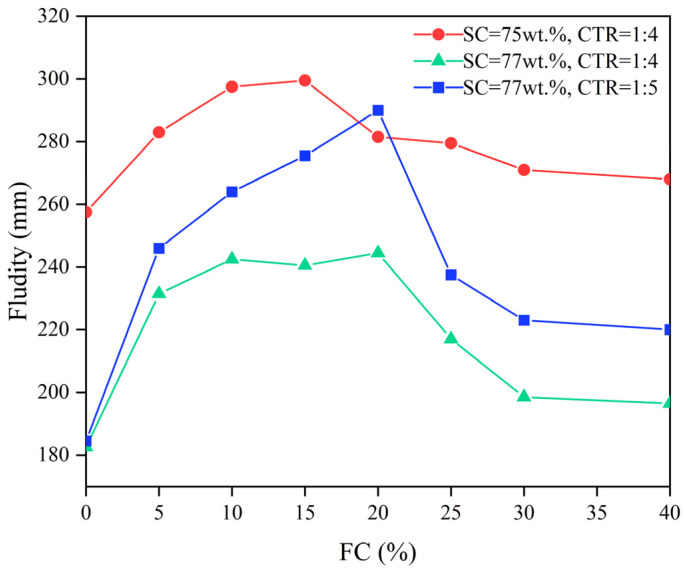
Relationship between the FC and the fluidity of FCPB.

**Figure 10 materials-15-07101-f010:**
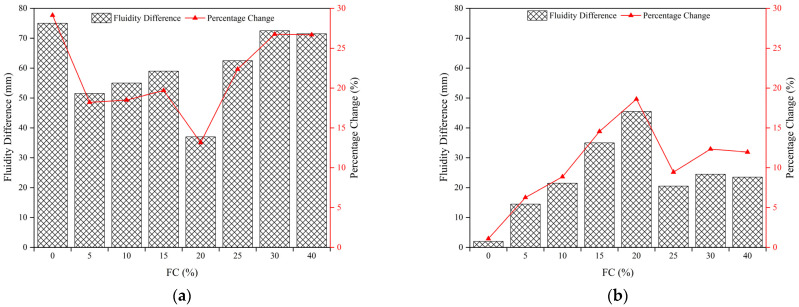
Relationship between the FC and the differences in the fluidity of FCPB: (**a**) CTR = 1:4 and (**b**) SC = 77 wt.%.

**Figure 11 materials-15-07101-f011:**
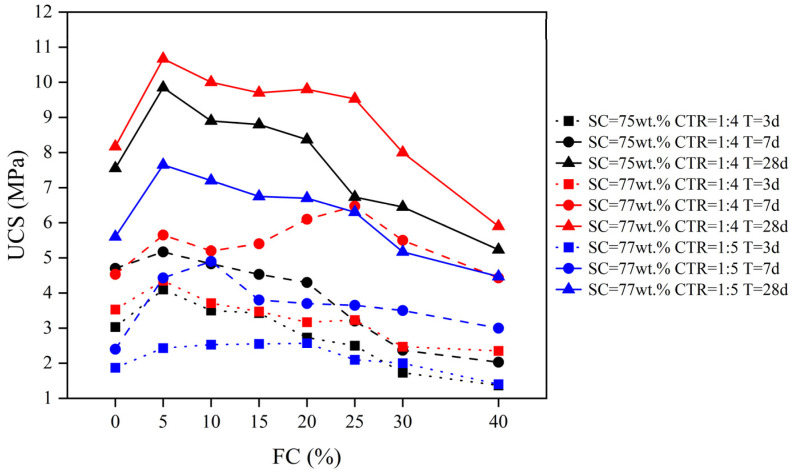
Relationship between the FC and the UCS of the FCPB.

**Figure 12 materials-15-07101-f012:**
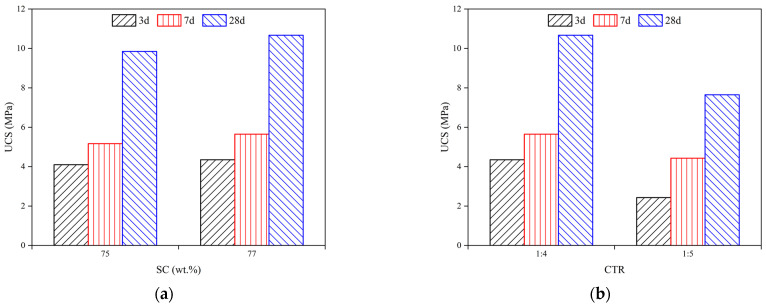
Relationship between the SC and the UCS of the FCPB (FC = 5%): (**a**) CTR = 1:4 and (**b**) SC = 77 wt.%.

**Figure 13 materials-15-07101-f013:**
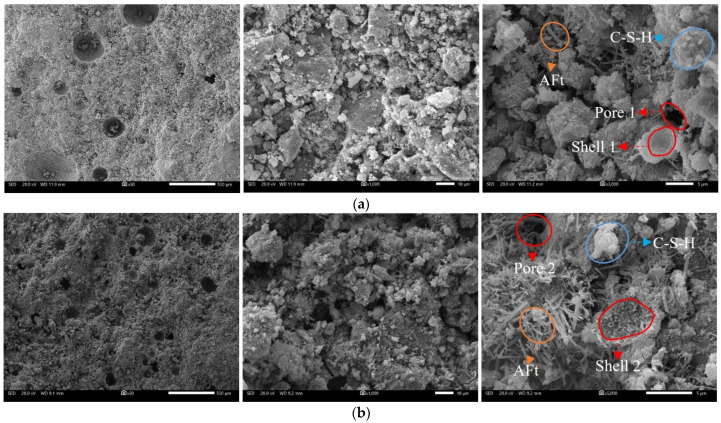
SEM images of an FCPB sample (SC = 77 wt.%, CTR = 1:5, and T = 28 d): (**a**) FC = 10% and (**b**) FC = 30%.

**Figure 14 materials-15-07101-f014:**
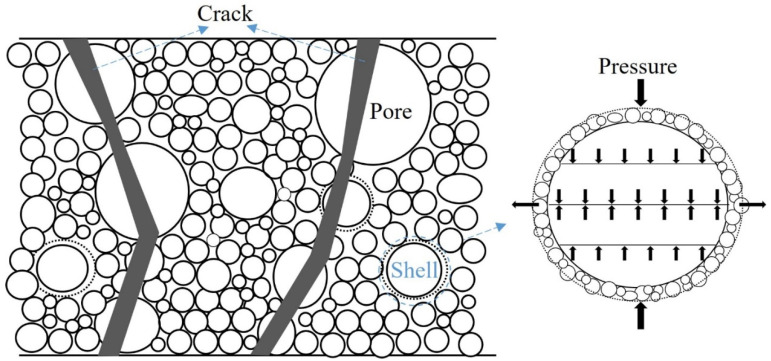
Microstructure of an FCPB sample.

**Table 1 materials-15-07101-t001:** Experimental design.

Scenario	Solid Content (wt.%)	Cement-Tailing Ratio	Curing Time (Days)	Foam Content (%)
1	75	1:4	3, 7, 28	0, 5, 10, 15, 20, 25, 30, 40
2	77	1:4	3, 7, 28	0, 5, 10, 15, 20, 25, 30, 40
3	77	1:5	3, 7, 28	0, 5, 10, 15, 20, 25, 30, 40

**Table 2 materials-15-07101-t002:** Physical properties of tailings.

Items	Wet Density (g/cm^3^)	Dry Density (g/cm^3^)	True Density (g/cm^3^)	Porosity (%)	Permeability Coefficient (cm/s)
Tailings	2.16	1.93	3.18	39.31	8.70 × 10^−4^

**Table 3 materials-15-07101-t003:** Chemical compositions of tailings.

Compositions	CaO	SiO_2_	Al_2_O_3_	Fe_2_O_3_	MgO	K_2_O	Other
wt.%	65.20	16.80	5.69	4.57	3.51	1.49	2.74

**Table 4 materials-15-07101-t004:** Min-Max normalization results of foaming ratio and half-life period of foam.

Foam Stabilizer Content (g/L)	0	0.5	1	1.5	2	3
Foaming Ratio	1.000	0.321	0.714	0.179	0.500	0.000
Half-life Period	0.000	0.466	0.309	0.362	0.640	1.000

**Table 5 materials-15-07101-t005:** Density of the FCPB slurry.

Scenario	SC (wt.%)	CTR	Foam Content							
			0%	5%	10%	15%	20%	25%	30%	40%
1	75	1:4	2088	2047	2030	1897	1883	1873	1730	1667
2	77	1:4	2153	2100	2023	2000	1977	1933	1863	1760
3	77	1:5	2077	2043	2020	1977	1933	1937	1853	1770

## Data Availability

Not applicable.

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
