# Peer review of "Investigation of Fluidity and Strength of Enhanced Foam-Cemented Paste Backfill"

_materials, 2022, doi:10.3390/ma15207101_

Round 1

Reviewer 1 Report

The purpose of this paper is to study some properties of enhanced foam-cemented paste backfill using industrial wastes. The topic is interesting and timely, since this subject has not yet been completely explored. Although the conclusions are rather expected, essentially supporting previous works. The experimental program is satisfactory, with an appropriate number of case studies and assessed properties. The overall structure of the paper is appropriate, but some suggestions are given. Figures and tables also require a few alterations. Regarding the abstract section, it shows a poor structure and the ideas are unclear. The English writing has also some issues, although not compromising the understanding of the content. More detailed comments are given below, but not exhaustively. As consequence, in my opinion this paper should only be accepted after major revision.

-          Some suggestions of English writing improvement are given in the attached pdf.

-          Title mentions mechanical properties; although, fluidity was studied and is not a mechanical property.

-          Abstract should be rewritten, specially the part referring to the conclusions. Authors give many details on the conclusions at a moment when readers are not properly aware of the content of the work. Hence, readers cannot completely understand the text. Authors should summarize the conclusions to essential information. Avoid too long sentences and clarify the ideas. The utilization of acronyms is driven to extreme and the text becomes very difficult to follow. Authors mention the study of compactness properties, but I could not found any work regrading this topic.

-          Table 1 and 2: include standards used to measure the properties; permeability coefficient of a powder is unusual

-          Clarify the amount of K12 (row 118), CMC (row 133) and FC (row 149). The percentage amount requires a base witch the proportion refers to (solid content?)

-          Row 136: definition of half-time period missing

-          Tables poorly numbered

-          Row 192-194: simplify writing, avoid repetition of expressions

-          Rows 203-206: this is supposed to be published knowledge, so it should be in the present tense and include references

-          Figure 7 and 8: remove limes between marks in the chart; eventually add correlation lines

-          Section 3.3.2: add text commenting the minimum difference for an intermediate FC (20%) for Figure 10 (a) and (b)

-          Rows 262-263: authors made copy/paste from rows 256-259, which makes the reading boring

-          Row 317: The authors state “when the foam was not added” in relation to images with FC=10% and 30%. Please clarify

-          Rows 320-326: add references

-          More detailed remarks are included in the attached pdf.

Reviewer 2 Report

The authors need to address the following comments and suggestions:

1.  English proof read the manuscript.

2. Reduce the use of abbreviations in the text and spell out most of the terms.

3. Define the wt%  (percent out of what) every time it is used.

4. Discuss the reasons for  changing a CMC content of 0.2 wt%.  How does  choosing a CMC content of 0.3 wt% impact the study resuts?

5.  How was the experimental design (solid content and foam content) defined? Does the experimental design cover all possible study variables?

Reviewer 3 Report

The authors investigated the mechanical and other properties of enhanced foam-cemented paste backfill. The works is interesting but there are several issues to address.

Abstract

The abbreviations SC and CTR were not defined at their first mention in the abstract

Materials And Methods

Materials

In this section only the materials used in this experiment should have been reported, but the authors have also reported the result which was supposed to be part of the RESULTS AND DISCUSSION section. The following statements ought to be in the RESULTS AND DISCUSSION section: “Figure 1 shows the particle size distribution of the tailings, which was 94 determined using laser particle size analysis technology. As can be seen, d10, d30, d50, 95 and d60 were 8.390 μm, 28.511 μm, 57.085μm, and 79.519 μm respectively, and the pro- 96 portion of tailings of <20 µm only reached 21.6%. The non-uniformity coefficient (Cu) of the tailing particles was 9.478 and the coefficient of curvature (Cc) was 1.218, indicating 98 the particle size gradation of tailings is good”.

The method used for determining the chemical composition of the tailings should be stated.

Foaming Agent Content

The numbers in the chemical formula “CH3(CH2)11OSO3Na” should be subscripted.

“The 0.1 wt.%, 0.2 wt.%, 0.3 wt.%, 0.4 wt.%, 0.5 wt.%, 0.6 wt.%, 0.7 wt.%, and 0.8 wt.% of K12 were measured out and dissolved in 100 ml of tap water, and then stirred at 1500 rpm for 3 minutes”. This is wrong. Did they prepare the stated concentrations in wt.% by dissolving a certain weight in 100 mL of water? They should be very clear on this.

Foam Stabilizer Content

“After determining the optimal foaming agent content, the 0.05 wt.%, 0.10 wt.%, 0.15 wt.%, 0.20 wt.%, and 0.30 wt.% of CMC were measured out and dissolved in 100 ml of  foaming agent solution, respectively”. This error is similar to the one pointed in the previous section. It should be rewritten to reflect the weight used per 100 mL of water. You cannot dissolve wt.% to get wt.%.

FCPB Preparation

There are two “Table 1”: “Physical properties of tailings” and Experimental Design”.

There are two “Table 2”: “Chemical compositions of tailings” and “Density of FCPB slurry”

Figure 3

The figure should further be described by using the letters (a), (b), (c), etc. on the images in a sequential order.

Conclusion

The significance of this study should be stated.

Round 2

Reviewer 1 Report

The authors have satisfactorily addressed most of my comments. Replace half-time by half-life in Table 4 (2 occurences).

Author Response

Thanks for the reminder, the authors have changed half-time to half-life.

Reviewer 2 Report

Accept in present form

Author Response

Thank you for your valuable comments!